# Bithiophene-Based Donor–π–Acceptor Compounds Exhibiting Aggregation-Induced Emission as Agents to Detect Hidden Fingerprints and Electrochromic Materials

**DOI:** 10.3390/molecules29235747

**Published:** 2024-12-05

**Authors:** Patrycja Filipek, Magdalena Kałkus, Agata Szlapa-Kula, Michał Filapek

**Affiliations:** Institute of Chemistry, Faculty of Science and Technology, University of Silesia, Szkolna 9, 40-007 Katowice, Poland; patrycja.filipek@us.edu.pl (P.F.); agata.szlapa-kula@us.edu.pl (A.S.-K.)

**Keywords:** aggregation-induced emission (AIE), latent fingerprints, donor–π–acceptor, electrochromism, luminescence

## Abstract

A group of bithiophenyl compounds comprising the cyanoacrylate moiety were designed and successfully synthesized. The optical, (spectro)electrochemical, and aggregation-induced emission properties were studied. DFT calculations were used to explain the reaction’s regioselectivity and to determine the molecules’ energy parameters (i.e., band gaps, HOMO levels, and LUMO levels). The aggregation-induced emission of compounds has been studied in the mixture of DMF (as a good solvent) and water (as a poor solvent), with different water fractions ranging from 0% to 99%. It has been shown that there are differences in the physicochemical properties of the obtained compounds due to the length of the alkyl chain in the ester group. Investigated derivatives were tested for their potential use in visualizing latent fingerprints and electrochromic materials.

## 1. Introduction

Fingerprints have two fundamental characteristics that allow for the easy identification of a person. These are uniqueness and immutability. These characteristics allow fingerprints to be used to identify criminals [1]. A fingerprint is a pattern of grooves and ridges (made up of sweat, oils, and sebum) left by contact between a finger and a surface. There are three types of fingerprint evidence: patent (those that have been transferred, for example, with blood or paint) [2], overt, and those that occur in most cases. In most cases, fingerprints are invisible and require tools to make them visible [3,4]. Most conventional LFP (latent fingerprint) visualization techniques rely on chemical reactions or interactions between developing substances and the components of the fingerprint residue, providing contrast between the surface and the fingerprint pattern [5]. Many LFP visualization techniques have been developed over the years. Examples of such techniques include powder spraying, cyanoacrylate (CA) glue fumes, ninhydrin (NH) spraying, and many other methods. These techniques are widely and eagerly used due to their ease of use, simplicity, and rapid results. However, like everything else, these techniques also have their drawbacks [6]. The most popular technique used to visualize LFPs is fingerprint powders. Powders can be made of many materials, such as carbon powder, starch, or silica gel. There are five types of powders: traditional, metallic, fluorescent, magnetic, and adhesive tape powders [7]. Recently, fluorescent powders have attracted the most significant interest. These powders exhibit excellent physicochemical properties (such as a high specific surface area, high adhesion, high contrast, high selectivity, and low background interference) [8,9,10]. Another example of a visualizing agent is cyanoacrylate glue. Cyanoacrylate fuming is a chemical method for detecting LFPs. It involves the anionic polymerization of CA (cyanoacrylate), which is probably initiated by various compounds in fingerprint residues, such as amino acids, fatty acids, and proteins [11]. This method provides a high degree of image acquisition but often requires further processing [12]. All of the previously mentioned methods have their advantages and disadvantages. The continuous development of existing techniques as well as the development of new ones are required.

As mentioned earlier, fluorescent powders are becoming increasingly desirable for LFP detection. At the same time, they are examples of combining known methods with a new approach that uses the AIE phenomenon to identify LFPs. The term AIE was first introduced in 2001 by Tang’s group. At that time, an abnormal phenomenon (according to the state of knowledge at the time) was observed, namely, a solution of 1-methyl-1,2,3,4,5-pentaphenylsilol in ethanol showed a significant increase in fluorescence after adding water. It was observed for the first time that aggregation can increase emissions rather than quench them [13]. Before 2001, it was believed that conventional chromophores showed efficient light emission only in a molecularly dispersed state, and the accumulation of molecules should be strictly avoided, which seriously hampered the practical applications of many leading chromophores [14]. Therefore, this phenomenon has revolutionized and opened many new applications [15]. Further experiments allowed us to determine the mechanism of this process, which is based on the restriction of intramolecular motion (RIM). Many AIE molecules contain molecular rotors, such as rotating aromatic rings. Once fully solvated, the rotation of the AIE molecules’ aromatic rotors absorbs the excited state’s energy, allowing for rapid energy decay without emission. After aggregation, intermolecular interactions between the AIE molecules restrict the rotation of the rotors, causing the molecules to decay via radiative channels. The RIR model explained the observed AIE in many molecular systems [16].

This combination of “two worlds” using the phenomenon of aggregation-enhanced emission in detecting latent fingerprints allowed us to contribute to the development of fingerprinting towards the use of fluorescent techniques. These methods allow for better contrast between the background and the trace. This method is developing rapidly, and several examples of AIE compounds are already used for LFP detection [17]. Ling D. and his group synthesized iridium Ir(III) complexes. As a result, the obtained compounds showed strong AIE properties and confirmed the mechanism of action mentioned earlier. Moreover, the LFP detection attempts were successful, and high image contrast was achieved [18]. Another example was presented by Xiaodong J. and his group, who developed AIE dyes using tetraphenylethene (TPE) or 1,1,2,3,4,5-hexaphenylsilol (HPS). As in the previous case, these dyes fulfilled their role and allowed LFP images to be obtained with good contrast on various porous and non-porous surfaces, and they also played an essential role in identifying latent traces [19]. Non-porous surfaces are smooth surfaces, such as displays, countertops, knives, and glasses [20], while porous surfaces are understood as all those that are not smooth, such as paper, cardboard, and coins [6]. In addition to the surface, many factors must be considered when developing a “detector” of latent fingerprints. In the powder method, it is important that the process of obtaining a given compound is simple and cheap and that its application does not damage the trace. In the spray method, using binary organic solvents or organic–water mixtures, especially organic solvents, inevitably causes damage to the LFP. Therefore, it is of importance to continue working in this field [21,22].

However, this work will focus on compounds whose core is based on bithiophenes. Bithiophene-based materials are among the most studied materials due to their extraordinary properties, such as low energy gaps, low oxidation potential, high thermal stability, and excellent photoelectric properties [23,24]. It is known that the electron-rich properties of thiophene rings provide many electrons for the entire system [25]. A cyano group is introduced to “improve” bithiophene-based structures. Introducing such a group enhances electron delocalization throughout the molecule via the conjugated skeleton. The planar structure of bithiophene promotes better conjugation, which is otherwise limited by the twisting of the end groups. Good electron delocalization produces equal C–C bond lengths in the thiophene units [26]. As a result, bithiophene derivatives with an attached cyano group exhibit a reduced energy gap, extended HOMO/LUMO separation, or a bathochromic shift in the longest absorption band spectra [27]. All of the features mentioned above are found in compounds based on bithiophene electrochromic materials. Chromic materials are materials that exhibit a reversible color change in response to an external stimulus, which can be temperature (specifically thermochromism) [28], light (specifically photochromism) [29], or oxidation and reduction after the application of electrical polarization. As a result, a color change occurs [30]. Several copolymers were developed, the core of which was based on bithiophene. These copolymers exhibited electrochromism, proving that bithiophene units have the most significant influence on this phenomenon [31,32].

Using such structures to identify hidden fingerprints (and not only as previously mentioned) yields very positive results. Visualization occurs on various surfaces; many details can be seen in the image. This is due to the tendency of such molecules (AIE nature) to have hydrophobic components in fingerprint residues. When AIE molecules interact with hydrophobic residues, they aggregate due to a hydrophobic effect, which reduces intramolecular motion and, consequently, improves their emission properties [33,34].

## 2. Results

The main goal of this research was to obtain donor–π–acceptor derivatives exhibiting the AIE phenomenon, showing electrochromism and/or increased affinity for latent fingerprints. Thus, based on our experiences described previously [35,36], aryl cyanoacrylate derivatives were synthesized as target systems. In this research, however, a strongly pi-excessive substituent was introduced into the structure of the molecule, i.e., bithiophenyl. The variable in this series of molecules was the alkyl substituent in the ester group. We assumed that the difference in the alkyl chain length will have a negligible effect on the electronic parameters of a single molecule but quite a significant effect on intermolecular interactions. Therefore, this should significantly impact the AIE phenomenon and its change within this series. The structures of all obtained molecules are shown in Figure 1 (below).

### 2.1. NMR Analysis

The studied derivatives were obtained by direct Knoevenagel-type condensation (Figure 2). An NMR analysis is extremely useful in this case because the substrate is an heteroaromatic aldehyde—the aldehyde hydrogen is characterized by a peak of about 9–10 ppm, i.e., a range in which there are few other groups (for this particular aldehyde, the signal appears at 9.8 ppm). As one can see in Appendix A, the reaction resulted in a quantitative conversion of the aryl substrate. On the other hand, the theoretical (main) products of such a reaction are E and Z isomers differing in the configuration of substituents at the newly formed double bond (Figure 2). The key to the analysis is the peak originating from vinyl hydrogen (–CH=C–). It is worth noting that already on the basis of the study of this peak, a shift is visible within the series (between 8.19 and 8.27), which means that there are some differences in the distribution of electron density. On the other hand, the reaction is highly selective—only the E isomer is formed. DFT calculations were found to be useful in explaining this issue (see Table 1). In the case of the formation of the E isomer, there is additional intramolecular stabilization–hydrogen bonding, i.e., an interaction between the oxygen of the ester group and the hydrogen located at carbon 4 of the bithiophene ring.

### 2.2. DFT Calculations

Theoretical studies are an excellent tool to help explain experimental data. Using Gaussian 16 software [37], quantum chemical calculations were performed for the presented compounds using the hybrid functional sets B3LYP and 6-311+G. The optimization of the structures was performed in a vacuum. Based on these geometries, the frontier orbitals of molecules in dichloromethane were calculated. The experimental dichloromethane solution was mimicked by using the polarizable continuum model [38,39] to obtain the UV–vis absorption spectra.

In the first step, the frontier orbitals were analyzed. The structures of the discussed derivatives are typical D-A systems, where the donor is a 2,2′-bitophene fragment and the acceptor group is a 2-cyanoacrylic ester. The HOMO orbital is mainly located on the donor part of the molecule and partially on the acceptor part. The LUMO orbital is located more on the part that has a greater tendency to withdraw electrons from electron-rich species (Table 1). Interestingly, the effect of changing the alkyl bond length at the 2-cyanoacrylic ester unit in the **M1**–**M4** derivatives is not significant. This is visible when comparing the ionization potential (IP) and electron affinity (EA) values. All molecules show similar IP values in the 6.11–6.15 eV range. The lowest value is characteristic of the **M3** derivative with a tert-butyl chain and the highest for P1-1 with a short methyl (Table 2). The situation is the same for the EA values, where the range is 3.01–3.08 eV. The calculated Eg(HOMO-LUMO) gaps confirm slight differences between HOMO and LUMO levels related to different alkyl bond lengths at the 2-cyanoacrylic ester unit. To estimate the energy barriers for the injection and transport rates of holes and electrons, the reorganization energies were calculated (Table 2). For the given compounds, both λ_hole_ and λ_electron_ are low, which indicates an efficient charge transport process [38,39]. Moreover, computational studies were carried out at the TD-DFT/B3LYP/6-311+G level (Appendix A). The calculations clearly indicate that the absorption band between 350 and 457 nm corresponds to the HOMO → LUMO transition. This transition is characterized by the greatest oscillator strength. Moreover, including the theoretical transition in the absorption band confirms the correct selection of calculation conditions.

### 2.3. Electrochemistry

The next stage of the research was a series of measurements using the cyclic voltammetry method, which is used to study the electrochemical properties of dissolved substances by measuring the electric current flowing through the electrochemical system in response to a linearly changing potential of the working electrode. This measurement allows for the ionization potential (IP), electron affinity (EA), and electrochemical band gap of the investigated compounds to be determined. The reduction and oxidation voltammograms obtained during the research are presented below (Figure 3).

In the case of reduction, the behavior of the molecules is typical for this type of derivative [35,36]. As one can see, a reduction is irreversible (from the thermodynamic point of view), with a distinct wave and almost identical peak onset values for all compounds, i.e., between −1.43 V and −1.47 V (see Table 3). This means that the electron density within the cyanoacrylate group is approximately similar, which is also consistent with the results predicted using the DFT calculations. In the case of oxidation, some differences can be noticed. For compounds with small-sized substituents (**M1** and **M2**), oxidation occurs when the potential is almost as high as 1 V (0.97 V and 0.99 V, respectively). Meanwhile, for derivatives **M3** and **M4**, this process occurs at 0.84 and 0.82 V. These differences directly affect the electrochemically determined energy gap, which is lower for the last two investigated compounds.

### 2.4. UV-Vis and Photoluminescence

In the next stage, measurements were performed using UV-vis and PL spectroscopy for diluted (c = 10^−5^ mol·dm^−3^) solutions in dichloromethane (Figure 4). When analyzing the obtained UV-vis spectra, it can be seen that all compounds tested exhibit maximum absorbance in a similar range of 410–420 nm. The highest molar absorption coefficient is observed for compound **M4**, i.e., a derivative with an octyl substituent. We also observe a slight hypsochromic shift, i.e., towards shorter wavelengths, caused by ester substituents (COOR) relative to the malononitrile derivative [35]. The tested compounds showed similar behavior when examining photoluminescence for dilute solutions. All of them emitted light at a wavelength of 493 nanometers with emission quantum yields between 0.7% (**M4**) and 3% (**M1**). The Stokes shifts are also almost equal (83–90 nm) for the entire series of investigated molecules (see Table 4).

### 2.5. UV-Vis Spectroelectrochemistry

Electrochemical studies of the discussed compounds have shown that they exhibit quasi-reversibility of both electrode processes (reduction and oxidation). Also, taking into account the commonly known electrochromic properties of bithiophenyl derivatives, a series of UV-vis spectroelectrochemical studies were performed (Figure 5; see also Appendix A). This method allowed us to observe the changes in the absorption of the tested derivative during its oxidation/reduction. The first basic requirement is, of course, that the individual formed in the redox process must be stable (not subject to degradation, decomposition, etc.). The second requirement is that the changes should occur in the spectroscopically accessible range, preferably (due to potential applications, e.g., as an electroactive layer in smart windows) in the visible light range. Our investigations have shown that all of the compounds meet these requirements. When analyzing the spectrum illustrating the behavior of the derivatives during oxidation, it is easy to see that there is a slight decrease in the intensity of the absorption peak, with the simultaneous appearance of a broad band lying between 500 and 900 nm (with an apparent maximum at 723 nm). This type of band appearing during oxidation is clear evidence that the charge is not localized in a single area of the molecule but is distracted within the molecule due to resonance stabilization (resulting from the presence of a conjugated system of double and single bonds). On the other hand, during the reduction in this group of compounds, a decrease in the original band can be observed with a simultaneous increase in an additional peak with a maximum of 340 nm, i.e., outside the visible range. In the case of the **M2** derivative, the compound solution is completely discolored, which means that after reducing the molecule, it does not absorb light from the visible range (it becomes transparent). It is also worth noting that in each case, after de-doping, the spectrum returned to its original state, confirming the reversible nature of electrode processes.

### 2.6. Aggregation-Induced Emmision Investigations

The AIE phenomenon studies began with finding the most beneficial good/bad solvent pair. A good solvent is, of course, one that provides high solubility of a given group of compounds (in this case, methanol, THF, MeCN, or DMF was tested), while a bad one causes precipitation of the solute (water or hexane). Of course, both solvents must be miscible with each other in any proportion. As can be seen in Figure 6, the **M1** derivative shows a practically constant QY (quantum yields) value at a water content in the range of 0–20% with a maximum at 510 nm. Then, an almost linear increase of up to 90% (QY = 68%) in efficiency as a function of fw is observed, with a new emission peak with a maximum at 460 nm, after which the emission efficiency is slightly lower. The **M2** derivative showed quite similar behavior. In this case, the efficiency starts to increase after exceeding fw = 30% with a maximum also at 510 nm and increases to reach a maximum for 80% (QY = 42%) with a new peak with a maximum at the same wavelength (460 nm). However, the difference will be most visible when we compare the above two compounds with derivatives M3 and M4. Molecule **M3** initially shows a decrease in QY. However, after exceeding fw = 30%, it increases to a maximum (450 nm with 5% QY) at a water content of 80% (see Appendix A). After exceeding this value, a new emission peak appears (at about 550 nm), and it begins to dominate the excitation of solutions with a higher water content. This means that the change in the ratio between solvents affects not only the degree of aggregation of molecules but also the type of aggregates formed. Similarly, for **M4**, initially, the intensity of the band with a maximum at 500 nm (7% QY) increases and reaches a maximum for fw = 50% (Appendix A). After exceeding this value, the band with a maximum at 570 nm begins to dominate the emission spectrum.

In all cases studied, aggregation caused the AIE effect, which is a desirable phenomenon for the compounds discussed (see also Figure 7). The results indicate that the dyes studied can be effective tools in forensics for visualizing latent fingerprints. Using a dye with the AIE effect provides more precise and higher contrasting images, which increases the accuracy and efficiency of the fingerprint identification process.

### 2.7. Identification of Latent Fingerprints

As part of the studies, an attempt was made to visualize a latent fingerprint. During the preliminary tests, photoluminescent properties of a thin layer of the tested compounds were tested. Under UV irradiation, all of the compounds showed intense yellow luminescence, confirming their potential to be used in fingerprint visualization applications (Figure 8). In the next step of the research, a fingerprint was imprinted on a glass plate, and then a tiny amount of dye was applied to this spot. A clear fingerprint was observed after exposing the plate to UV radiation yielding the yellow fluorescence.

As mentioned earlier, compounds exhibiting aggregation-enhanced emission (AIE) are excellent candidates for latent fingerprint (LFP) detection due to their hydrophobic nature and optical properties. Above is a summary of LFP detection for three different subsoils for four compounds, **M1**–**M4**, under UV light (As can be seen at first glance, the best surface turned out to be metal, whereas for all four compounds, it was possible to obtain an image with partially visible minutiae (a close-up picture of minutiae is provided in Figure 9). However, the image for compound **M1** is the best due to it having the most visible minutiae. Looking at the second substrate, which is plastic, only the image for compound M4 is the most promising due to the visible confirmation of contact with the surface and very few minutiae, which cannot be observed in the case of the other compounds. The last subsoil used was ceramic. In this case, compounds **M1**, **M2**, and **M4** strongly interacted with the substrate, causing a lack of background separation from the image, which can only confirm contact with the subsoil without more information (see Figure 10).

In summary, as a result of the comparison of four compounds (different substituents at the cyanoester group) on three different surfaces, it can be stated that the compound containing an octyl (**M4**) attached to the cyanoester group is a more “universal” compound than the other one. It allows for good visualization on two different surfaces. However, to obtain an image containing more details, therefore being more desirable, compound M1 presents itself better here, the application of which is limited to metal surfaces.

## 3. Materials and Methods

### 3.1. General Methods

All chemicals and starting materials were commercially available and were used without further purification. Solvents were distilled as per the standard methods and purged with nitrogen before use. All reactions and measurements were carried out under an argon atmosphere unless otherwise indicated. Column chromatography was carried out on Merck Silica Gel 60 (Merck Millipore, Burlington, MA, USA). Thin-layer chromatography (TLC) was performed on silica gel (Merck TLC Silica Gel 60 F254) (Merck Millipore, Burlington, MA, USA). ^1^H NMR and ^13^C NMR spectra were recorded using a Bruker Avance UltraShield 400 MHz spectrometer (Bruker, Karlsruhe, Germany). The peaks were referenced to the residual CDCl_3_ (7.28 and 77.04 ppm) resonances in 1H and ^13^C NMR spectra, respectively. UV/Vis spectra were recorded with a Hewlett–Packard model 8453 UV/Vis spectrophotometer in dichloromethane solution (Agilent Technologies, Santa Clara, CA, USA). Electrochemical measurements were carried out with an Eco Chemie Autolab PGSTAT128n (Metrohm AG, Ionenstrasse, 9100 Herisau, Switzerland) potentiostat using glassy carbon (with diam. 2 mm) as a working electrode, while platinum coil and silver wire were used as auxiliary and reference electrodes, respectively. Potentials are referenced with respect to ferrocene (Fc), which was used as the internal standard. Cyclic voltammetry experiments were conducted in a standard one-compartment cell, in dichloromethane (Carlo Erba, Emmendingen, Germany, HPLC grade), under argon. Bu_4_NPF_6_ (Aldrich, St. Louis, MO, USA; 0.2 M, 99%) was used as the supporting electrolyte. UV-vis spectroelectrochemical measurements were performed in a 1 cm quartz cuvette with Indium Tin Oxide (ITO, with 10 Ω per square) as a glass working electrode; platinum and silver wire were used as auxiliary and reference electrodes, respectively. The quantum theoretical calculations were performed with the use of the density functional theory (DFT), with an exchange–correlation hybrid functional B_3_LYP [40,41,42,43] and the basis 6-311+G [44,45,46] for all atoms. The calculations were carried out with the use of the Gaussian16 program.

### 3.2. Synthesis and Characterization

#### 3.2.1. Synthesis of **M1**, Methyl 2-cyano-3-(2,2′-bithiophen-5-yl)prop-2-enoate



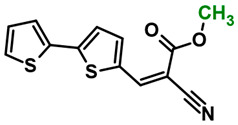



In a 25 mL flask, 0.51 mmol (100 mg) of aldehyde was placed, and then 0.51 mmol (50 mg) of methyl cyanoacetate was added. The whole mixture was dissolved in 1 mL of ethanol under argon. During argonation, 1 mL of triethylamine was added. An orange-yellow precipitate fell out immediately. Then, another 1 mL of ethanol was added to dissolve the precipitate and stirred for 24 h at room temperature. The precipitate was filtered and rinsed with hexane. No further column was required. The reaction yield was 87%. ^1^H NMR (500 MHz, CDCl_3_) δ 8.29 (s, 1H), 7.69 (d, *J* = 4.0 Hz, 1H), 7.41 (dd, *J* = 7.8, 2.6 Hz, 2H), 7.28 (d, *J* = 4.1 Hz, 1H), 7.11 (dd, *J* = 5.1, 3.7 Hz, 1H), 3.93 (s, 3H). ^13^C NMR (126 MHz, CDCl_3_) δ 163.45, 147.73, 146.58, 139.27, 135.74, 134.19, 128.54, 127.53, 126.54, 124.50, 116.01, 97.31, 53.22.

#### 3.2.2. Synthesis of **M2**, Ethyl 2-cyano-3-(2,2′-bithiophen-5-yl)prop-2-enoate



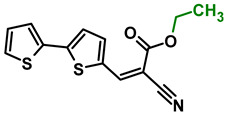



In a 25 mL flask, 0.51 mmol (100 mg) of aldehyde was placed, and then 0.51 mmol (60 mg) of ethyl cyanoacetate was added.The whole mixture was dissolved in 1 mL of ethanol and argonated. During argonation, 1 mL of triethylamine was added. An orange-yellow precipitate fell out immediately. Then, another 1 mL of ethanol was added to dissolve the precipitate and stirred for 24 h at room temperature. The precipitate was filtered and rinsed with hexane. No further column was required. The reaction yield was 86%. ^1^H NMR (500 MHz, CDCl_3_) δ 8.27 (s, 1H), 7.67 (d, *J* = 4.0 Hz, 1H), 7.42–7.38 (m, 2H), 7.26 (d, *J* = 4.0 Hz, 1H), 7.10 (dd, *J* = 5.0, 3.7 Hz, 1H), 4.38 (q, *J* = 7.2 Hz, 2H), 1.40 (t, *J* = 7.1 Hz, 3H). ^13^C NMR (126 MHz, CDCl_3_) δ 162.93, 147.51, 146.33, 139.06, 135.78, 134.27, 128.52, 127.44, 126.48, 124.48, 116.02, 97.84, 62.48, 14.24.

#### 3.2.3. Synthesis of **M3**, Tert-butyl 2-cyano-3-(2,2′-bithiophen-5-yl)prop-2-enoate



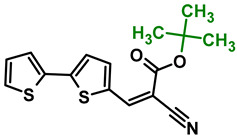



In a 25 mL flask, 0.51 mmol (100 mg) of aldehyde was placed, and then 0.51 mmol (71 mg) of tert-butyl cyanoacetate was added. The whole mixture was dissolved in 2 mL of ethanol and argonated. During argonation, 1 mL of triethylamine was added. The solution remained clear (dark yellow). The whole mixture was stirred for 24 h at room temperature. The precipitate (needles) formed was filtered off and rinsed with hexane. No further column was required. The reaction yield was 65%. ^1^H NMR (500 MHz, CDCl_3_) δ 8.16 (s, 1H), 7.62 (d, *J* = 4.0 Hz, 1H), 7.37–7.35 (m, 2H), 7.22 (d, *J* = 4.0 Hz, 1H), 7.06 (dd, *J* = 4.9, 3.9 Hz, 1H), 1.58 (s, 11H). ^13^C NMR (126 MHz, CDCl_3_) δ 161.71, 146.87, 145.45, 138.60, 135.84, 134.36, 128.49, 127.29, 126.33, 124.40, 116.20, 99.60, 83.40, 28.03.

#### 3.2.4. Synthesis of **M4**, Octyl 2-cyano-3-(2,2′-bithiophen-5-yl)prop-2-enoate



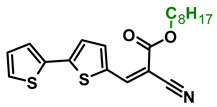



In a 25 mL flask, 0.51 mmol (100 mg) of aldehyde was placed, and then 0.51 mmol (100 mg) of octyl cyanoacetate was added. The whole mixture was dissolved in 2 mL of ethanol and argonated. During argonation, 1 mL of triethylamine was added. After a while, a light yellow precipitate fell out of the clear yellow solution. The whole solution was stirred for 24 h at room temperature. The resulting precipitate was filtered and rinsed with hexane. A column in dichloromethane was carried out to isolate the product residue in the filtrate. The reaction yield was 46%. ^1^H NMR (500 MHz, CDCl_3_) δ 8.28 (s, *J* = 6.3 Hz, 1H), 7.69 (d, *J* = 4.0 Hz, 1H), 7.52 (d, *J* = 4.0 Hz, 1H), 7.45 (dd, *J* = 3.7, 0.9 Hz, 1H), 7.42 (dd, *J* = 3.7, 0.9 Hz, 1H), 7.24 (ddd, *J* = 4.9, 3.7, 1.0 Hz, 1H), 7.11 (dd, *J* = 5.0, 3.8 Hz, 2H), 1.85–1.68 (m, 5H), 1.36 (s, *J* = 3.4 Hz, 5H), 0.94–0.87 (m, 11H), 0.84 (t, *J* = 7.2 Hz, 3H) ^13^C NMR (126 MHz, CDCl_3_) δ 162.90, 147.41, 146.17, 139.13, 135.76, 134.25, 128.52, 127.46, 126.45, 124.43, 115.94, 97.80, 66.56, 31.80, 29.21, 29.18, 28.57, 25.83, 22.67, 14.14.

## 4. Conclusions

During the described research, four donor–π–acceptor derivatives were designed and obtained. Moiety bithiophenyl was used as the donor, while cyanoacrylate esters were used as the acceptor. DFT calculations were used to explain the reaction’s regioselectivity and to determine the molecules’ energy parameters (i.e., band gaps, HOMO levels, and LUMO levels). It has been shown that there are differences in the physicochemical properties of the obtained compounds due to the length of the alkyl chain in the ester group. Electrochemical investigations revealed that oxidation is easier for compounds with longer alkyl chains. Spectroelectrochemical measurements were performed; they confirmed the electrochromic properties of the tested derivatives in the visible light range. AIE properties in DMF:water mixtures were also conducted. Considering all the tested optical, (spectro)electrochemical, and electrochromic properties, it can be stated that the tested compounds have a high potential for use in electroactive devices (for example, in smart windows). On the other hand, high fluorescence intensity, electrochemical stability, and the ability to enhance emission in the aggregated state make them excellent candidates for fingerprint visualization, offering clear and reliable images, which is crucial for precise identification in forensic investigations.

## Figures and Tables

**Figure 1 molecules-29-05747-f001:**
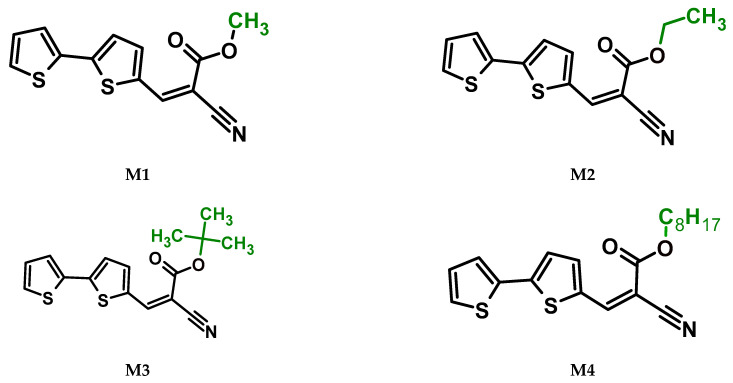
Structures of obtained compounds (**M1***–***M4**).

**Figure 2 molecules-29-05747-f002:**
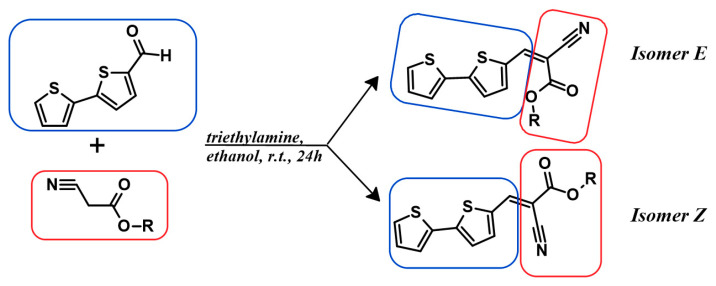
A sketch of the synthetic pathway with two potential reaction products.

**Figure 3 molecules-29-05747-f003:**
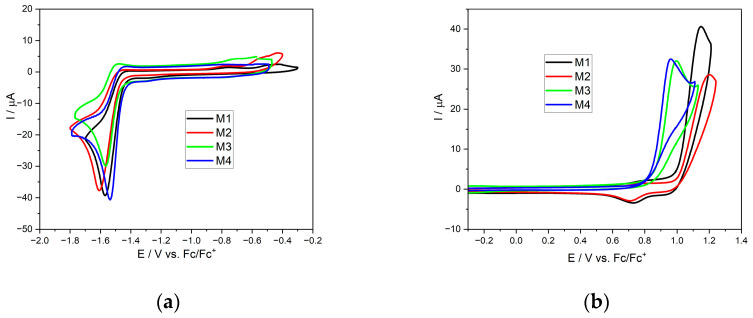
Cyclic voltammograms of investigated compounds with sweep rate of ν = 100 mV/s, 0.1 M Bu_4_NPF_6_ in dichloromethane. (**a**) During negative potential sweeping. (**b**) During positive potential sweeping.

**Figure 4 molecules-29-05747-f004:**
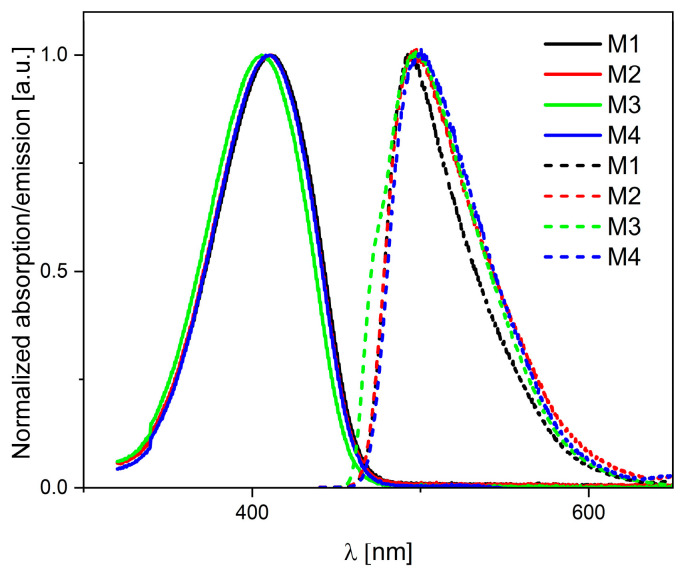
Normalized electronic absorption in dichloromethane (solid lines) and PL spectra of investigated compounds (dotted lines); c = 1 × 10^−5^ mol/L.

**Figure 5 molecules-29-05747-f005:**
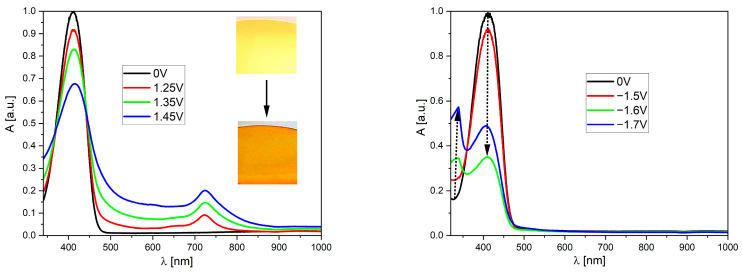
The UV-vis spectroelectrochemistry of the studied derivatives in the dichlorometane solution (c = 1 × 10^−5^ mol/L, as an inset on each graph, all potentials vs. the Fc/Fc^+^ redox couple). Selected spectra were recorded during the oxidation of the **M1** derivative (**left**), and there was a reduction in the **M2** derivative (**right**). As the inset in the spectrum, there are pictures of the compound in neutral form and after oxidation. The dotted arrow indicates changes in peak intensity.

**Figure 6 molecules-29-05747-f006:**
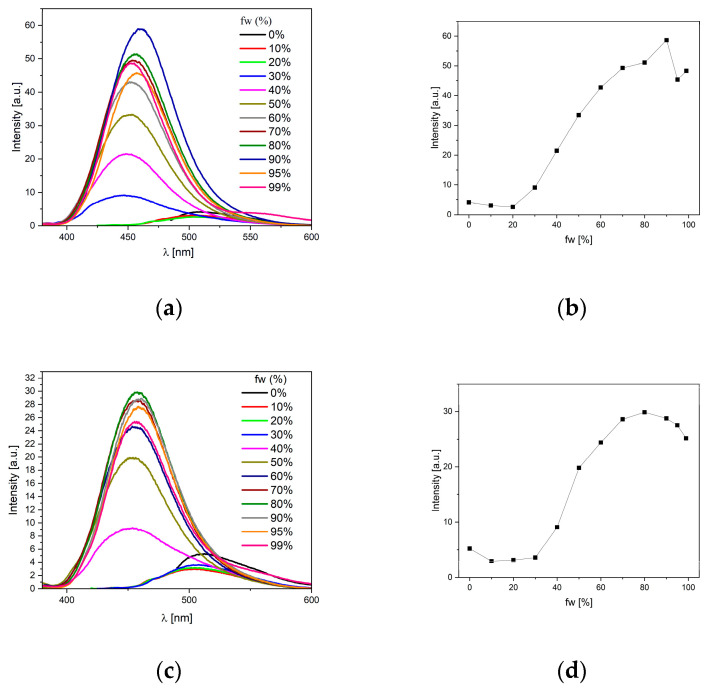
Emission spectra (**a**,**c**) and fw vs. fluorescence plot (**b**,**d**) for **M1** (top) and **M2** (bottom) in mixtures of DMF and water (c = 1 × 10^−5^ mol/L).

**Figure 7 molecules-29-05747-f007:**
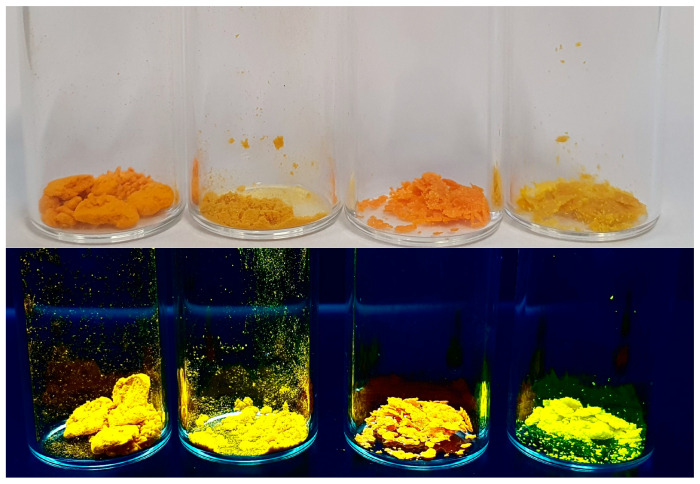
Pictures of investigated compounds (from left, **M1**, **M2**, **M3**, and **M4**) in daylight (**top**) and during 366 nm light irradiation (**bottom**).

**Figure 8 molecules-29-05747-f008:**
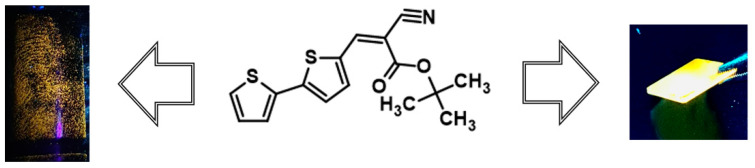
A visualization of latent fingerprints (**left**) and an example of the photoluminescence of the compound (**middle**) in the form of a thin film (**right**).

**Figure 9 molecules-29-05747-f009:**
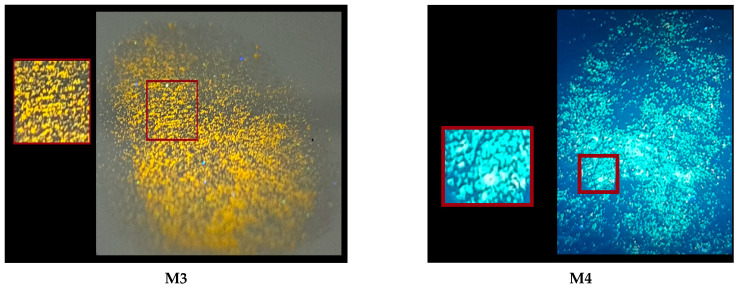
Fluorescent images of the LFPs.

**Figure 10 molecules-29-05747-f010:**
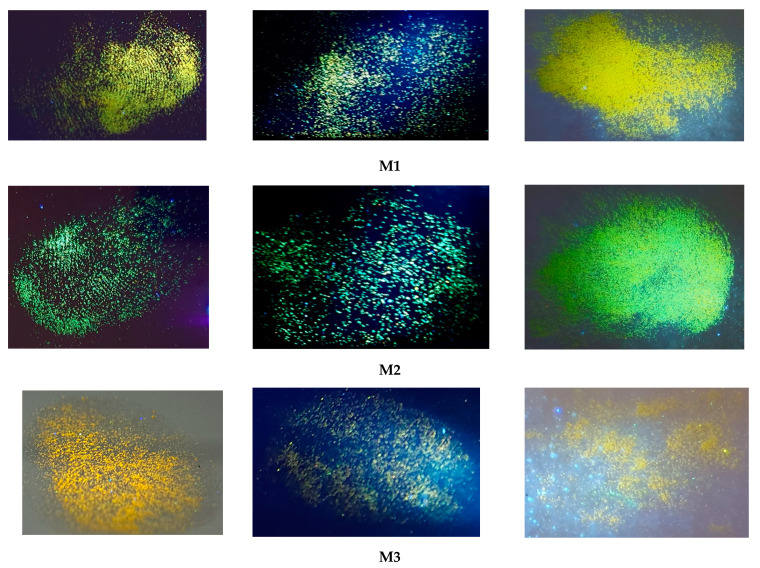
Fluorescent images of the LFPs on different surfaces.

**Table 1 molecules-29-05747-t001:** The HOMO and LUMO levels together with the calculated bond lengths.

CODE	HOMO	LUMO	BOND LENGTHS
**M1**	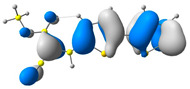	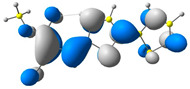	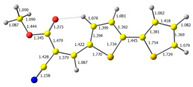
	E = −6.15 eV	E = −3.08 eV	
**M2**	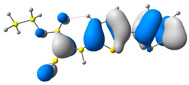	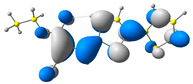	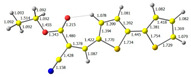
	E = −6.14 eV	E = −3.07 eV	
**M3**	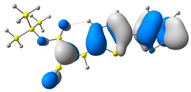	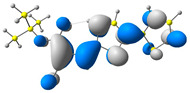	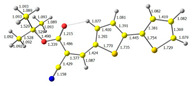
	E = −6.11 eV	E = −3.01 eV	
**M4**	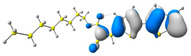	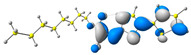	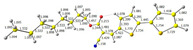
	E = −6.14 eV	E = −3.06 eV	

**Table 2 molecules-29-05747-t002:** Calculated IP and EA, energy gap, and hole and electron reorganization energies and extraction potentials for **M1**–**M4**.

Code	IP(a) [eV]	IP(v) [eV]	EA(v) [eV]	EA(a) [eV]	λ_hole_ [eV]	λ_electron_ [eV]	HEP [eV]	EEP [eV]	Eg (HOMO − LUMO)(a) [eV]
**M1**	5.99	6.15	3.08	3.25	0.28	0.31	5.87	3.40	3.07
**M2**	5.98	6.14	3.07	3.23	0.28	0.32	5.86	3.38	3.08
**M3**	5.96	6.11	3.01	3.18	0.28	0.33	5.83	3.34	3.10
**M4**	5.98	6.14	3.06	3.22	0.29	0.32	5.85	3.38	3.09
EEP = E^0^(M^−^) − E^−^(M^−^); HEP = E^+^(M^+^) − E^0^(M^+^); λ_electron_ = EEp − EAv; λ_hole_ = IP_v_ − HEP; ^a^ adiabatic; ^v^ vertical.									

**Table 3 molecules-29-05747-t003:** IP, EA, and energy gap values for **M1**–**M4**.

Code	Eox	Ered	IP ^a^	EA ^a^	Eg (CV) ^b^
**M1**	0.97	−1.44	−6.07	−3.66	2.41
**M2**	0.99	−1.47	−6.09	−3.63	2.46
**M3**	0.84	−1.46	−5.94	−3.64	2.30
**M4**	0.82	−1.43	−5.82	−3.67	2.25

^a^ calculated from CV measurements (IP = −5.1 − Eox; EA = −5.1 − Ered; ^b^ Eg(CV) = Eox (onset) − Ered (onset)).

**Table 4 molecules-29-05747-t004:** Photophysical parameters obtained from UV-vis spectroscopy and photoluminescence measurements.

Code	λ_max_(Em)	λ_max_(A)	λ_onset_	Eg (opt)	Stokes Shift [nm]
**M1**	493	410	463	2.68	83
**M2**	496	410	462	2.68	86
**M3**	496	406	457	2.71	90
**M4**	498	409	461	2.69	89

## Data Availability

Data Availability Statement:The original contributions presented in this study are included in the article/Appendix A. Further inquiries can be directed to the corresponding author.

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
