# Peer review of "Bithiophene-Based Donor–π–Acceptor Compounds Exhibiting Aggregation-Induced Emission as Agents to Detect Hidden Fingerprints and Electrochromic Materials"

_molecules, 2024, doi:10.3390/molecules29235747_

Round 1
Reviewer 1 Report
Comments and Suggestions for Authors
In this manuscript, Patrycja Filipek et al. synthesize and characterize a group of bithiophenyl compounds comprising the cyanoacrylate moiety and explain the regioselectivity of the reaction and determine the energy parameters of the molecule (i.e., band gap, HOMO, and LUMO levels) by DFT calculations. The authors have investigated the optical and spectroelectrochemical properties of compounds. The emission of aggregation-induced compounds in mixtures with different water fraction ranges is carefully studied. And it is proved that the length of the alkyl chain in the ester group of the compound is different, and the physicochemical properties of the response are also distinct. Finally, the application of materials in fingerprint visualization is demonstrated. It is hoped that the author can systematically supplement the results of electrochromic-related tests and characterizations. Overall, it is worth publication once the following issues are addressed.
Major Issues:
1) Since the title of this article includes the study of electrochromism, we have not seen the relevant test and characterization results of the electrochromic system in the text, and hope that the author will carry out supplementary tests in the future.
2) On line 201-211 of page seven, the whole description is not particularly accurate, it is recommended to reorganize the language to carefully argue, for example, 1.43 V may lack a negative sign, and why the onset potential is similar, the electron cloud density is almost the same, please carefully argue or cite relevant literature to prove.
3) In Tables 2 and 3, the Eg values trends of M1, M2, M3 and M4 are theoretically calculated to be opposite to the measured trends, please explain the relevant reasons.
4) On page nine, the analysis is inconsistent with the phenomenon in Figure 4. Please demonstrate why M4 has the highest molar absorption coefficient and what is the significance.
5) On line 222-224 of page ten, based on the results of the electrochromic tests presented in this paper, we believe that these materials cannot be used as electrochromic active layers in smart windows, so it is recommended to dissolve the materials in a gel electrolyte to prepare an integrated electrochromic device, and then systematically study its electrochromic properties and take photos of the actual electrochromic products.
6) In Figure 6, it is inconsistent with the data analysis, and needs to be rechecked and corrected, and it needs to be reformatted due to lack of clarity.
7) The application of M1 and M2 materials in identification of latent fingerprints is not found in this paper, and whether the relevant experimental conclusions can be given.
Minor Issues:
1) On page five, Spectrum 1 is proposed to be changed to Figure 1 and placed in the supplementary material file, the NMR spectra of M1 and M2 materials are not found in the text, and it is recommended to complete them
2) On line 175 of page six, it is advisable to explain the meaning of the IP and EA abbreviation when it first appears.
3) In Figure 7, the picture needs to be improved, and it is recommended that the compound be placed in a vial and taken uniformly.
Typos and others:
1) Please check the punctuation marks of the Eg values of M3 and M4 in Table 4.
2) On page nine, please double-check the reference format in this sentence. “We also observe a slight hypsochromic shift, i.e., towards shorter wavelengths, caused by ester substituents (COOR) relative to the malononitrile derivative [D&P]”.
Comments on the Quality of English LanguageThe Quality of English Language need to be improved.
Reviewer 2 Report
Comments and Suggestions for Authors
The presented text introduces new compounds for fingerprint detection. The studies used reasonable methods to prove the compounds' potential. However, I recommend that the article be accepted after revision. Some points need to be clarified (see comments and questions).
Comments:
2) Line 160: You have to cite the functional and basis sets! It is better to cite everything in the experimental section. However, it is written that calculations were done in Gaussian 09, but in line 159, Gaussian 16 is mentioned. I need some clarification.
3) line 185: Please remove space in "TD-DFT/ B3LYP/6-311+G" to "TD-DFT/B3LYP/6-311+G". I checked the SI, but only one transition is compared with the experiment. Can you add all transitions with calculated spectra to SI?
4) Line 202: The reduction is irreversible. There is no evidence of a quasi-reversible process. I believe the measurements were performed at room temperature, but I would suggest measuring at a low temperature (-77°C).
5) line 203: missing "-" before number 1.43 V
6) Use dichloromethane in all text.
7) Spectroelectrochemistry: I suggest changing the title to "UV-Vis spectroelectrochemistry." Several types of spectroelectrochemistry exist—IR, UV-Vis, etc. You do not need to explain the requirement for spectroelectrochemistry.
8) Figure 5: Please clarify where oxidation occurs and where reduction.
Questions:
1) Which solvation model did you use to model the dichloromethane environment? I need help finding this information.
2) What setup did you use for spectroelectrochemical measurements? Which cell did you use? More information about spectroelectrochemistry needs to be provided in the experimental section! I can only comment a spectroelectrochemistry in more detail with this information.
3) Why did you use overpotential for oxidation in Figure 5?
4) Did you measure SEC for oxidized and reduced species for all compounds? I can see only M1 upon reduction or oxidation. Why are you presenting only reduction for M2 and oxidation for M4? What about M3?
5) Why did you not use DCM for AIE studies when DCM was used in all previous cases?
6) Why are you not presenting DPV results? DPV is mentioned in the experimental part.
Round 2
Reviewer 1 Report
Comments and Suggestions for Authors
This revision is better than the last manuscript. But some small mistakes, which I requested, have not been corrected. Please check the punctuation marks of the Eg values of M3 and M4 in Table 4. It should be “2.71 and 2.69”, not “2,71 and 2,69”. As a result, I suggest that the manuscript can be accepted after a double check carefully.
Reviewer 2 Report
Comments and Suggestions for Authors
I recommend accepting this manuscript for publication in its current form.